# Can We Identify Subgroups of Patients with Chronic Low Back Pain Based on Motor Variability? A Systematic Scoping Review

**Lars Dijk [1,2]**[ID]**, Marika T. Leving [2], Michiel F. Reneman [3] and Claudine J. C. Lamoth [1,*]**

1 University of Groningen, University Medical Center Groningen, Department of Human Movement Sciences, 9713 AV Groningen, The Netherlands; larsdijk1998@gmail.com
2 General Practitioners Research Institute, 9713 GH Groningen, The Netherlands; marika@gpri.nl
3 University of Groningen, University Medical Center Groningen, Department of Rehabilitation Medicine, 9713 AV Groningen, The Netherlands; m.f.reneman@umcg.nl
* Correspondence: c.j.c.lamoth@umcg.nl; Tel.: +31-(0)50-361-6047

**Abstract:** The identification of homogeneous subgroups of patients with chronic low back pain (CLBP), based on distinct patterns of motor control, could support the tailoring of therapy and improve the effectiveness of rehabilitation. The purpose of this review was (1) to assess if there are differences in motor variability between patients with CLBP and pain-free controls, as well as inter-individually among patients with CLBP, during the performance of functional tasks; and (2) to examine the relationship between motor variability and CLBP across time. A literature search was conducted on the electronic databases Pubmed, EMBASE, and Web of Science, including papers published any time up to September 2021. Two reviewers independently screened the search results, assessed the risk of bias, and extracted the data. Twenty-two cross-sectional and three longitudinal studies investigating motor variability during functional tasks were examined. There are differences in motor variability between patients with CLBP and pain-free controls during the performance of functional tasks, albeit with discrepant results between tasks and among studies. The longitudinal studies revealed the persistence of motor control changes following interventions, but the relationship between changes in motor variability and reduction in pain intensity was inconclusive. Based on the reviewed literature, no stratification of homogeneous subgroups into distinct patterns of motor variability in the CLBP population could be made. Studies diverged in methodologies and theoretical frameworks and in metrics used to assess and interpret motor variability. In the future, more large-sample studies, including longitudinal designs, are needed, with standardized metrics that quantify motor variability to fill the identified evidence gaps.

**Keywords:** lower back; chronic pain; EMG; kinematics; motor control; motor variability

## 1. Introduction

Chronic low back pain (CLBP) is a highly disabling condition, which calls for effective interventions that target pain, disability, and functioning [1–3]. Although improvements for guidelines of CLBP clinical practice have been achieved over the years [4], the effectiveness of therapies, including rehabilitation, remains modest at best [5,6]. Variations in treatment outcomes could be driven by the heterogeneity of the CLBP population [7–11]. In CLBP, central and peripheral nociceptive processes are influenced by multidimensional factors [12–14]. The identification of homogeneous subgroups of patients with CLBP, based on distinct patterns of motor control, could support the tailoring of therapy and improve the effectiveness of rehabilitation [15,16].

Differences in the patterns of motor control can be identified using outcome measures based on muscle activation or kinematic movement patterns, representing the outcomes of neural structures and processes [17]. Studies have shown aberrant alterations in trunk-pelvis coordination and in muscle recruitment during functional tasks in patients with CLBP compared to pain-free controls, such as during walking [18]. While there is a

long tradition of studying movements based on mean values and ensemble averages of movement patterns in CLBP over the last decades, the concept of motor variability has received considerable attention in recent years [19–21]. From a motor-control perspective, features of motor variability provide insight into stable and adaptive motor control [22–24]. One of the origins of motor variability is 'motor redundancy', defined as having more degrees of freedom (DOFs) than necessary to execute a motor task [25]. Therefore, each repetition of a movement is based on variations in neural and motor patterns, while leading to the same goal. Hence, movement variability is inherent across multiple executions of the same task, in kinematic, kinetic, and muscle activation patterns, and can therefore be regarded as a natural phenomenon.

The interpretation of systematic differences in motor variability is challenging because it is unknown whether these reflect the adaptability of the neuromotor system and should therefore be considered beneficial, or reflect changes in physiology as a result of pathological processes [26]. For instance, trunk stiffness reduces movement variability and could lead to increased spinal loading while moving [27] but could also reflect an attempt to stabilize the spine [21]. High movement variability could be a sign of reduced neuromuscular control but might be beneficial for the re-distribution of joint loads, muscle activity, and ligament stress [28]. Thus, differences in motor variability may be both a cause and effect of pain and might also contribute to the chronicity of CLBP [29].

At present, the source of changes in motor variability is not known. It could be the direct effect of CLBP or could be mediated by fear of movement, disabilities, and/or the consequences of CLBP. Furthermore, the heterogenous CLBP population might present with different patterns of motor variability and even be classified in homogenous subgroups based on distinct patterns of motor variability. Therefore, the primary aims of this systematic scoping review were to (1) assess if there are differences in motor variability between patients with CLBP and pain-free controls, as well as inter-individually among patients with CLBP, during the performance of functional tasks; and (2) examine the relationship between motor variability and CLBP across time.

## 2. Materials and Methods

### 2.1. Study Selection

A systematic scoping review protocol was conducted (registration prior to data extraction on 29 August 2021: https://osf.io/JGC69/). This systematic scoping review is reported according to the PRIMSA-ScR guidelines [30].

For inclusion in the review, studies should assess adults (>18 years) with CLBP (>3 months, without the identification of underlying pathologies) and include measures of motor variability, operationalized as the muscle recruitment patterns measured using electromyographic (EMG) signals (thoracolumbar region) and/or kinematic movement patterns during functional tasks. Studies were excluded if patients were assessed with acute/recurrent LBP, with experimentally induced pain, without a clear definition of the CLBP, with underlying pathologies, or if DOFs were artificially reduced in the study protocol (the vast majority of studies on motor control in CLBP involve controlled laboratory tasks, such as isolated bending tasks with pelvic stabilization [31]), did not report measures of motor variability, were not written in English, or were identified as one of the following types: review, meta-analysis, single-case studies, and study protocols.

### 2.2. Search Strategy

Three electronic databases (PubMed, EMBASE, Web of Science) were searched without limitations on time (final search: 16 September 2021. Search strategy: Supplementary Materials S1) using a broad search strategy based on relevant medical subject heading (MeSH) terms. Reference lists of included articles were hand-searched. Two reviewers (L.D. and J.B.) independently identified potentially relevant articles based on titles and abstracts. Full-text articles were retrieved and checked independently for compliance with

inclusion and exclusion criteria. Disagreements were resolved by consensus or a third reviewer (M.T.L.).

### 2.3. Quality Assessment

The quality of the evidence was independently evaluated by two authors (L.D. and J.B.) with a modified version of the Downs and Black checklist [32,33] (Supplementary Materials S2 ans S3). This scoping review examined non-randomized studies; therefore, items 8, 9, 13, 14, 17, 19, 23, and 24 were removed. The modified tool consisted of 19 items, adapted to the relevant outcome measures. Disagreements were resolved by consensus or a third author (ML). Study quality was classified as high ($\geq$75%), moderate (50–74%), or low (<50%). All studies were included in the data synthesis irrespective of quality.

### 2.4. Data Extraction

Extracted data included author(s) and year of publication, study design, sample characteristics (e.g., sex, age, weight, stature, BMI, duration since first symptoms, sample size), clinical questionnaires, functional tasks, measures of motor variability, and main findings. If reported, effect sizes were extracted (Supplementary Materials S4. The data was not appropriate for pooling; therefore, a visual summary of the outcomes of the cross-sectional studies is provided, to give an overview of the evidence (gaps) in the current literature).

## 3. Results

### 3.1. Characteristics of the Included Studies

A total of 401 articles were retrieved from the online databases. A flow diagram of the study selection process is presented in Figure 1. The search yielded 22 cross-sectional studies comprising the following functional tasks; bending, gait, lifting, standing, sitting, and sit-to-stand (STS). Three longitudinal studies examined the within-subject relationship of motor variability and CLBP over time. Overall, 397 patients with CLBP and 357 controls were included.

### 3.2. Risk of Bias

A full overview of the methodological quality assessment of included studies is presented in Supplementary Materials S3. Total quality scores ranged from 66.7% to 86.7%. Fourteen studies were classified as high quality ($\geq$75%) and 11 studies were classified as moderate quality (50–74%). Overall, the studies scored poorly on the questions concerning blinding (item 15) and power calculations (Item 27).

### 3.3. Differences in Motor Variability between Patients with CLBP and Controls
3.3.1. Bending

Three studies examined 3D kinematic trunk patterns during repeated trunk flexion-extension tasks in patients with CLBP with low pain intensity (VAS < 2) and controls (Table S1) [34–36]. Patients exhibited higher local trunk stability than controls over multiple cycles (4–10), as quantified by a lower maximal Lyapunov exponent ($\lambda$max; $p = 0.03$) for patients with CLBP than controls. The $\lambda$max describes the capability to resist local perturbations [37], with lower values implying higher local stability of trunk movements [34]. A Goal Equivalent Manifold (GEM) analysis revealed the equal performance of a goal-directed repeated trunk flexion-extension task (i.e., maintaining constant movement time). GEM motor performance was quantified by decomposing the body-level variability in non-goal-equivalent variability (causing deviation from the task) and goal-equivalent variability (no effect on the task performance). No significant differences between the groups were observed for the goal-equivalent variability, non-goal-equivalent variability, and the ratio of both, implying that CLBP did not affect performance and flexibility during dynamic trunk movements [35]. The variability of coordination patterns, expressed by a relative phase (RP) analysis of sagittal lumbar-pelvis coupling, was influenced by task demands

(velocity/symmetry). Although no significant main group effect on the deviation phase (DP, variability in inter-segmental coordination) values of lumbar-pelvis rotations was observed, the CLBP group demonstrated more coordination stability (less adaptability) of the lumbar-pelvis coupling during highly demanding trunk flexion-extension compared to controls [36]. The adaptability of trunk flexion-extension movement patterns was similar between groups during a goal-directed task [35], but patients with CLBP demonstrated higher local stability of trunk movements [34] with less adaptability during more demanding tasks [36].

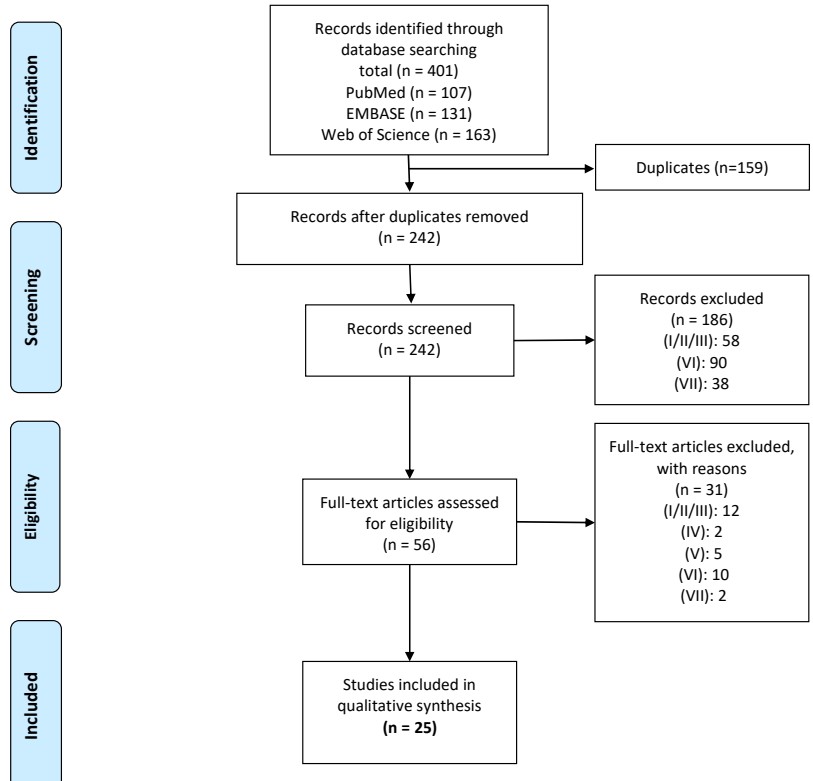

**Figure 1.** Flow diagram of study inclusion. Legend: (I) included people with acute/recurrent LBP; (II) experimentally induced pain; (III) without a clear definition of CLBP; (IV) reported that subjects had underlying pathologies; (V) study protocols where degrees of freedom were artificially reduced; (VI) did not report measures of variability; or (VII) were identified as on the following publication types: review, meta-analysis, single-case studies, and study protocols.

### 3.3.2. Gait

Gait outcome measures were related to spatiotemporal gait parameters [38–40], kinematic patterns of the trunk [38,41–47] and lower extremity [42], and bilateral erector spinae (ES) EMG patterns [44,45] (Table S2). Regarding the spatiotemporal gait parameters, the results are inconclusive. A significantly lower stride length variability (standard deviation) is reported in patients with CLBP than in controls during attentional demanding (Stroop) tasks [38]. Other studies reported higher stride time and length variability during both single and dual-task walking [39] and a higher coefficient of variation (CV) of stride time in a task with experimentally diminished visual feedback in patients with CLBP compared to controls [40]. With respect to coordination, the repeatability of the relative movement between trunk-pelvis during gait was lower in patients with CLBP compared to controls. The stride-to-stride variability (CV of the ensemble average for a single subject over repeated strides) of angular trunk-pelvis displacements in the frontal, transverse, and sagittal planes were lower ($p \leq 0.05$) in CLBP patients [41]. A more 'in-phase' coordination of sagittal trunk-pelvis movements was evident over the stance and swing phase [42]. A cognitive

dual-task provoked larger variability of stride-to-stride trunk movements in patients with CLBP but not in controls [43]. High attentional demanding dual-tasks decreased the variability of the RP of thoracic and pelvic rotations more in patients with CLBP compared to controls [38].

The adaptations of trunk movements to walking speed perturbations were different in patients with CLBP compared to controls. Patients had higher RP variability of frontal plane thoracic-pelvic rotations and lower RP variability of transverse thoracic-pelvic and lumbar-pelvic rotations compared to controls [44,45]. Similar results were found using a Principal Component Analysis (PCA). After the extraction of the global pattern of movement using PCA, the variability of residual patterns of thoracic and lumbar rotations was lower in the transverse and enhanced in the frontal plane, particularly at higher walking speeds [44]. High Pearson correlation coefficients (r) were reported between pelvis and thorax residual transversal rotations (r = 0.93 at 0.61 m/s, r = 0.81 at 1.61 m/s in CLBP compared to r = 0.70 at 0.61 m/s, r = 0.49 at 1.61 m/s in controls) [46]. On the contrary, a higher ($p < 0.001$) CV of frontal, sagittal, and transverse rotational movement patterns of the pelvis was evident in CLBP, pointing towards increased movement irregularities [47]. A lower deviation phase (SD of the ensemble RP curve; $p < 0.05$) for sagittal trunk-pelvis and pelvis-thigh coordination revealed less variable inter-segmental phase relation during walking in patients with CLBP compared to controls [42].

Patients with CLBP revealed poorly coordinated lumbar ES (LES) activity during velocity perturbations when walking on a treadmill (incrementally increased [45] or unexpected (faster/slower) changes up to 6.2 km/h [44]). Using PCA, the 'global invariant' muscle activation patterns were separated from the variable residual pattern. Bilateral LES global pattern variability was lower in patients with CLBP compared to controls, with higher residual pattern variability at higher walking velocities. The latter implies alterations in the LES patterns in terms of irregular phase shifts, amplitude modifications, and additional bursts [44]. For velocities higher than 2.2 km/h, the first extracted principal components of the PCA covered less variance in CLBP ($25 \pm 3\%$) compared to controls ($35 \pm 6\%$) and this trend progressed with increasing velocities [45].

Overall, gait studies indicate the presence of a lower variability of trunk movements during gait in CLBP, particularly in the transverse plane; a higher variability in the frontal plane and in movements of the pelvis; and inconclusive results with respect to outcomes that quantify stride- and step-related variability. With respect to muscle activity patterns, the lower variability of trunk movements was accompanied by less variability of global patterns of muscle activity and higher residual muscle activation pattern variability.

### 3.3.3. Lifting

Three studies examined the variability during repetitive lifting tasks in terms of 3D kinematics of the spine, hip, and lower extremities [48]; angular movements of the spine [49]; and muscle activity of the LES [50] (Table S3). A lifting-induced fatigue task revealed the higher local stability (lower λmax) of angular hip rotations in the frontal ($p < 0.05$) and transverse ($p < 0.01$) planes in patients with CLBP compared to controls, indicating a different control strategy for the hip in patients [48]. However, late-fatigue conditions (cycles 20–40) were associated with unstable (higher λmax values) lifting-lowering in both groups for the spine flexion-extension and angular hip movements ($p < 0.05$). No group differences were observed for the λmax of angular movements of the knee, ankle, and spine in both early- (initial 20 cycles) and late-fatigue conditions. With a metronomically-tuned lifting task, no group differences were observed in the magnitude of task-related spinal angular (sagittal) movements (the angular offsets and angular variability). However, a Recurrence Quantification Analysis, a method to address movement variability, showed that the structure of the variability of spinal movements differed between patients with CLBP and controls. For direct task-related angular trajectories, the percentage of determinism (%DET) was high for both groups (99.8%), implying high repeatability of the lifting task. In contrast, the %DET of accessory angular trajectories (i.e., movement variability not directly

related to the task execution) was higher in CLBP in 8 out of 12 locations along the spine ($p < 0.05$), indicating that the structure of the variability was more deterministic compared to controls [49]. In similar repetitive lifting tasks, with more repetitions (>25 cycles), the distribution of the root mean square of LES activation of eight locations increased to more caudal in controls, whereas, in patients with CLBP, no redistribution of activity to different regions was seen, but there was an overall increase in EMG amplitude (i.e., same LES regions active over the duration of the task) [50].

### 3.3.4. Standing, Sitting, and Sit-to-Stand (STS)

During standing on a rigid surface, the dynamic properties of anterior–posterior (AP) CoP time-series in patients with CLBP revealed more regularity, with lower sample entropy [51] and larger dimensionality [51,52] in CLBP than in controls. By increasing cognitive task difficulty, patients with CLBP decreased the percentage of recurrence (regularity) and trend (stationarity) of CoP time-series, though not to the same extent as controls [52]. Conflictingly, no differences in the predictability (approximate entropy) of CoP trajectories were reported when exposed to a postural control task with perturbations, suggesting predictable movement patterns in both groups [53].

Two studies analyzed the same STS task with different analytical approaches, respectively: a trial-to-trial SD of segment angles and a PCA [54], followed by an uncontrolled manifold approach (UCM) to assess joint coordination [55]. Based on a PCA, principal components (PC) were extracted that accounted for >90% of the variance of the seven limb angles in the sagittal plane: ankle, knee, hip, lumbosacral joint, cervical spine, elbow, and shoulder [54]. A lower number of PCs was suggested to represent stronger coupling between joint angles and fewer variable movements. In patients with CLBP, two PCs explained >90% of the variance, while in controls, three PCs were extracted on the condition that the center of mass (CoM) had the highest distance from the base of support. This was interpreted as reduced flexibility in the most unstable phase of the task. However, increasing the task difficulty (narrow surface with closed eyes) did not result in differences between the groups [54]. Across 15 trials, the mean SD of head and trunk angles were higher ($p < 0.05$) in patients with CLBP, suggesting that patients stabilized the CoM and head position through increasing joint configuration variability more than controls [54]. The UCM approach distinguished a component stabilizing the CoM and head positions (performance variables) leading to task performance flexibility (UCM variability: $V_{ucm}$) and a component destabilizing the task performance (orthogonal variability: $V_{ort}$). A significantly greater $V_{ort}$ of the horizontal head positions and a lesser $V_{ucm}$ of horizontal CoM positions indicated less flexible motor coordination strategies in patients with CLBP compared to controls [55]. Postural coordination during an arm-raising task in a sitting position was less variable in patients with CLBP than in controls. A decreased variability of anticipatory postural adjustment onset latencies in self-initiated, voluntary movements was observed, as well as a significantly lower SD of onset latencies of the bilateral internal oblique relative to the deltoid muscle onset in patients with CLBP compared to controls [28].

### 3.4. Longitudinal Interventions: Chronic Low Back Pain and Motor Variability over Time

Motor variability and clinical variables were reported in three longitudinal intervention studies that examined the effects of core stabilization exercises (CSE) [56], the motor control training (MCT) of repeated isolated voluntary transversus abdominis (TrA) contractions [57], and sensorimotor training (SMT) [58] (Table S5).

At the baseline of the CSE intervention, patients with CLBP exhibited trunk displacement ranges in all planes during gait which were similar to controls. However, the CV of the relative movement between trunk-pelvis (repeatability) during gait was lower in the CLBP group. After six weeks of CSE training, the CV of trunk-pelvis kinematics in the frontal and transverse planes was increased significantly in the CLBP group [56]. The CV sagittal trunk-pelvis kinematics remained lower in the CLBP group after six weeks of CSE. The CSE training decreased pain ($p < 0.001$) and functional disability scores ($p < 0.001$).

MCT, at baseline and at two weeks, led to a reduction ($p < 0.05$) in the CV of TrA activation in patients with CLBP during gait. Effects on TrA activity were retained at a six-month follow-up [57]. Lower scores on the visual analogue scale for pain ($p = 0.0047$) and improvements in Patient-Specific Functional Scale scores ($p < 0.02$) were reported. However, weak relationships between the CV of TrA and VAS ($r = -0.17$) and PSFS ($r = -0.071$) scales were reported [57].

The kinematic data of a postural control task showed no significant changes after the four-week SMT intervention in the ApEn of the AP CoP time series (i.e., no change in predictability). Moreover, the ratio of task-specific (UCM analysis; the amount of variability returning the CoM to its steady-state) and task-deviating joint angle variability (causing unwanted change) remained unchanged, implying no significant effect of SMT on joint angle variability. A significant ($p < 0.001$) improvement on the Oswestry Disability Index was found in the SMT group at a four-week follow-up (12 points, but not in the control group, which showed a four-point improvement). Overall, the studies demonstrated the persistence of motor control changes following interventions. However, the causal relationships between changes in movement variability and pain and functional capacities could not be proven in the CLBP population.

## 4. Discussion

### 4.1. Findings

The results of our scoping review demonstrated that there are differences in motor variability between patients with CLBP and pain-free controls during the performance of functional tasks, albeit with some discrepant results between tasks and among studies. Temporal relationships between changes in movement variability and pain and functional capacity were not evident in the longitudinal studies.

The present scoping review focused on studies that examined motor variability in functional tasks instead of isolated, controlled tasks. Studying isolated movements is required to test biomechanical models and gain insight into mechanically related processes that contribute to the development and course of CLBP, such as intervertebral movements [59]. When interested in motor variability, however, functional tasks are considered where the DOFs within a task are minimally restricted, because task demands will influence the motor strategies used to complete the task [60]. The results of the reviewed studies demonstrated that among functional tasks there were differences in motor variability between patients with CLBP and pain-free individuals, but these differences were not consistent. Figure 2 shows that, overall, there were studies that reported high and low variabilities in motor control parameters. To interpret the heterogeneity of the results, it is important to take into account the theoretical framework and associated metrics adopted by the included studies.

### 4.2. Metrics of Motor Variability

Several studies addressed variability with metrics that provide information about the magnitude of the variability such as RMS, CV, or SD (e.g., of step length, range of motion, EMG) neglecting time-dependent fluctuations and structure in these fluctuations. These conventional metrics fit into a conceptual framework in which motor variability is considered 'noise', and low trial-to-trial variability with a low SD reflects consistent behavior, whereas a higher SD might point towards a lack of control [61]. Over the last years, however, there is a growing interest in within-person and/or intraindividual variability from a dynamical systems perspective, which regards variability as an inherent property of human movement and provides a window to study the coordination of system components as well as its stable and adaptive features. From this perspective, human motor behavior is considered intrinsically variable both within and between individuals, and an in-depth analysis of variability, including both its temporal as well as spatial characteristics, may provide insight into the underlying control structures and mechanisms related to movement impairments [21,62–64].

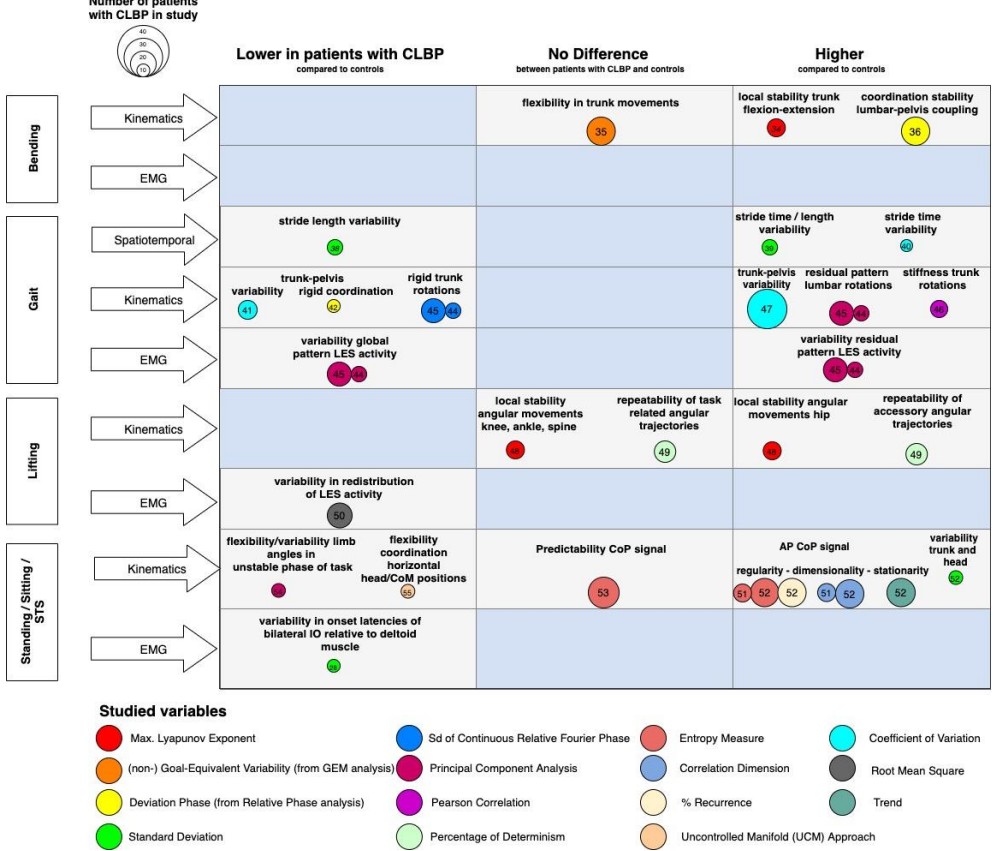

**Figure 2.** Findings and literature gap map of the cross-sectional studies. Legend: Differences in motor variability between patients with CLBP and controls (*x*-axis) across various functional tasks and levels of measurement (*y*-axis). The bubbles represent the methodology (in terms of used metrics) and their size represents the number of included patients with CLBP in the study. The numbers in the bubbles represent the references. Knowledge gaps are marked as blue fields. Abbreviations: LES: lumbar erector spinae; CoM: Center of Mass; CoP: Center of Pressure; AP: anterior-posterior; IO: internal oblique.

Most included studies examined motor variability using concepts and tools derived from dynamical systems theory, using numerous computational forms to characterize properties of motor variability such as correlation-based measures (PCA, detrended fluctuations analysis), pattern stability methods (PCA and RP analysis), and predictability-related measures (Recurrence Analysis, λmax, sample entropy, and UCM approach) (Figure 2 Supplementary Materials S4). Although all of them address movement dynamics and recognize the importance of movement variability, they are conceptually different. For instance, the λmax quantifies the system's ability to resist 'natural' local perturbations, and weak fluctuations are suggested to indicate local stability that can be steadily maintained. The rate of decay of small perturbations is referred to as local stability [65]; a system might be locally stable but globally cannot resist larger disturbances. Coordination dynamics describe and predict changes in patterns of oscillating body segments. In patients with CLBP, the variability of trunk-pelvis RP is considered an index of pattern stability [21]. Similarly, a PCA was used to extract the (in-)variant patterns of segment coordination from the more variable components, as an index of flexibility in kinematics and EMG [45]. The UCM theorem examines multi-joint coordination in relation to multi-DOF for task performance. It partitions variability as essential and invariant to the task and the variability that affects it [66]. The UCM provides information about the flexibility of task performance, which was reported to be lower in patients with CLBP [55]. The λmax, coordination stability, and UMC might thus reflect different motor control characteristics. For instance, fatigue during a lifting task is associated with a higher λmax, while higher pattern stability (expressed

by a lower mean and SD of RP) in patients with CLBP in gait is reported. An implication of the use of the general term motor variability for the present review is that the metrics could not be pooled together. Therefore, a meta-analysis of the effect sizes to generalize the (inconclusive) results between studies could not be carried out.

### 4.3. Heterogeneity in Motor Variability

When focusing on the discrepant results among studies and between tasks, the variety in functional tasks and task-specificity, per se, does provide different behavioral repertoires for the human body as a complex system to solve the motor tasks, quantifying a range of movement options inherent to motor variability. Moreover, differences in task constraints may lead to different motor variabilities; during dual-tasking, patients with CLBP might choose different priorities, for instance, maintaining cognitive performance at the expense of the motor task during gait [43] or vice versa [38]. The motor solutions available for a given task might serve various goals, such as the stabilization of the spine, maintaining the CoM within the base of support, end-point accuracy, or mitigating fear-avoidance beliefs [53]. Hence, it is crucial to consider the main goal of the task when analyzing motor variability. This was not consistent across the studies in our scoping review and ranged, for instance, from stabilizing performance variables (UCM) during STS [55] to maintaining constant movement time (GEM) during bending [35].

The substantial between-subject variance in motor variability in CLBP could also be driven by the heterogeneity of the CLBP population phenotyping [7–11]. Biopsychosocial models suggest that the fear of pain keeps a tight rein on motor behavior in patients with CLBP [67]. Specific movement adaptations, aimed at minimizing pain or the fear of pain, could be founded through reinforcement learning [20]. Some functional tasks might not have been threatening enough by individual participants to evoke a protective motor strategy, or motor strategies could be influenced by depressive symptoms, catastrophizing, and somatic anxiety [68]. A substantial amount of patients with CLBP present increased responsiveness of nociceptive neurons in the central nervous system to their normal or subthreshold afferent input, a phenomenon known as central sensitization [69]. Because central sensitization affects the central nervous system, it is assumed that functioning is related to central sensitization and pain. Nonetheless, the relationship between central sensitization and motor functioning—or, more specifically, motor variability—has not been extensively investigated.

It is widely recognized that CLBP presents as a multidimensional problem [70]. Remarkably, we found that the studies that addressed motor control and variability scarcely reported the clinical and demographic characteristics of the study sample. Multifactorial models are needed that include clinical and demographic factors to better understand differences in motor variability.

### 4.4. Pain Intensity and Patterns of Motor Variability

Proposed contrasts in motor variability between acute and chronic pain suggest that motor adaptations vary across time and might change when the pain is relieved [71]. Longitudinal studies [56–58] observed lower pain intensity and improved functional capacity in patients with CLBP after interventions. In studies with relatively high pain intensity at baseline (VAS 5.55 $\pm$ 1.7 [56], VAS > 4 [57]), motor variability shifted towards the levels of controls in the follow-up phase; contrastingly, no changes in motor variability (the predictability of the CoP sway) were evident in the intervention group with relatively low pain intensity at baseline (VAS 2.5 $\pm$ 0.7) [58]. A plausible reason for the reduced pain intensity and conflicting results regarding motor variability might be that specific muscles are responsible for pain alleviation and improved neuromuscular coordination, and that there might be a relationship between motor variability, muscle de-conditioning, movement direction, and the type of exercise performed [72].

### 4.5. Changes in Motor Variability in Patients with CLBP: Pathology or Genius Adaptation?

Optimal adaptive variability could possibly have two ends of the spectrum: variability beyond the upper limit, implying an unstable system that is sensitive to perturbations, and variability below the lower limit, implying a stereotypical system without exploratory behavior, which is, therefore, less capable of adapting to perturbations [71]. This paradigm is manifested in a recently published theory, which proposed the existence of two phenotypes in LBP: 'tight' control and 'loose' control over trunk movements [73]. However, the generalization of patients with CLBP during functional tasks and targeting treatments to these phenotypes requires caution. The results of this review indicate that the theory should be elaborated given considerations to the whole range of multidimensional CLBP conditions. The individual-specific motor behaviors attributed to beliefs and attitudes towards pain and the multidimensional clinical representation of CLBP should be covered. Furthermore, the context- and task-specificity of movements might give rise to specific intraindividual motor strategies, shifting the control strategies across the spectrum.

### 4.6. Limitations and Recommendations for the Future

As can be appreciated from Figure 2, a central issue within the literature regarding motor variability in CLBP is the enormous variety in motor variability metrics. Future studies about motor variability should formulate clear conceptual frameworks and hypotheses around motor variability and use consistent metrics. Inconsistent findings between studies may also arise due to differences in normalization methods, (subtle) changes in functional tasks, and differences in pain intensity or fear-avoidance behaviors of patients with CLBP prior to the study. Clinical parameters should be reported more comprehensively in order to compare the findings to other studies. The majority of the studies had cross-sectional study designs ($n$ = 22) with small sample sizes ($n$ = 20–55), therefore the stratification of homogeneous sub-groups remains indefinite and, either way, lacks statistical power. Only three longitudinal studies with a maximum of 15 participants in the intervention group were found [56]. Assumptions on whether changes in patterns of motor variability were either a cause or an effect of the CLBP are therefore indefinite. Future research should focus on longitudinal designs with multiple measurement points such as single-case experimental designs [74,75], which potentially yield new insights into the causality between adaptations in motor variability in the development of, and during, sustained CLBP.

### 4.7. Clinical Implications

The results of the present review show that it is not viable to identify homogeneous subgroups of patients with CLBP based on motor variability alone. Longitudinal studies, however, indicate that there is the potential for interventions targeting motor variability. Clinically, the results emphasize the need for the development and study of personalized approaches, and single-subject-like designs, to examine interventions.

## 5. Conclusions

The sample heterogeneity in terms of motor variability within patients with CLBP was observed. Substantial differences in the used methodologies and metrics and small sample sizes in studies did not allow us to stratify subgroups of patients with CLBP based on motor variability. Future studies should clearly operationalize motor variability and associated metrics, focus on longitudinal single-case experimental designs, and report patients' clinical phenotyping more comprehensively. The results of the few longitudinal studies revealed the potential for interventions to influence motor control changes and decrease pain intensity across time. However, a clear relationship between motor variability and CLBP was not evident, and the type of intervention differed widely.

**Supplementary Materials:** The following are available online at https://www.mdpi.com/article/10.3390/biomechanics1030030/s1, Supplementary Materials S1: Search Strategy; Supplementary Mate-

rials S2: Quality Assessment; Supplementary Materials S3: Methodological Quality; Supplementary Materials S4, Tables S1–S5: Characteristics of the included studies.

**Author Contributions:** Conceptualization, L.D., M.T.L. and C.J.C.L.; methodology, L.D., M.T.L., M.F.R. and C.J.C.L.; data curation, L.D., M.T.L., M.F.R. and C.J.C.L.; writing—original draft preparation, L.D.; writing—review and editing, M.T.L., M.F.R. and C.J.C.L.; visualization, L.D.; project administration, L.D., M.T.L., M.F.R. and C.J.C.L. All authors have read and agreed to the published version of the manuscript.

**Funding:** This research received no external funding.

**Institutional Review Board Statement:** Not applicable.

**Informed Consent Statement:** Not applicable.

**Data Availability Statement:** All available data is included in this manuscript.

**Acknowledgments:** The authors acknowledge Jelmer Braaksma for his contribution to the data extraction and quality assessment of this manuscript.

**Conflicts of Interest:** The authors declare no conflict of interest.

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
