# Peer review of "Can We Identify Subgroups of Patients with Chronic Low Back Pain Based on Motor Variability? A Systematic Scoping Review"

_2673-7078, doi:10.3390/biomechanics1030030_

Round 1

Reviewer 1 Report

This literature study shows a thorough very systematic analysis of research papers in the field of Chronic Low Back Pain (CLBP) to evaluate any difference in motor variability during the performance of functional tasks, further investigating the association between motor variability and CLBP over time.

In general, the review is of interest to a fairly wide audience. In particular, this overview and analysis will help other researchers in the field to provide clearer hypotheses around motor variability using consistent metrics and preferably employing longitudinal studies.

Twenty-five articles out of 401 reviewed were selected for this review. In this regard, I think that integration is necessary to better understand the impact and motives of exclusion. So, please, add the reason for exclusion for the full-text assessed retrieves in the flow chart.

Finally, here are my last minor remarks:

Data extraction

- line 111: not dealing with social but rather bio-medical aspects, it is better to replace "gender" with "sex".

- line 111: the term “length” seems very dubious from an anthropometric point of view. Body length means the vertex-planta measurement in the supine position. In all other cases, the correct term is stature.

Appendix D

-modify the "length" term as above and possibly express it in cm instead of m.

Figure 1

-I suggest writing roman numerals in capital letters.

Figure 2

-It is unclear how the sizes of the bubbles in Figure 2 were scaled. Subjectively?

Reviewer 2 Report

(row 127-131)

In the description of the criteria marked as (i) you can find criterion vii, which does not appear in Figure 1. Maybe it is worth removing them?

Reviewer 3 Report

Comments to the authors manuscript ID biomechanics-1418243 entitled “Can we identify Subgroups of Patients with Chronic Low Back Pain based on Motor Variability? A Systematic Scoping Review”. This manuscript is a review about Chronic low back pain.

The manuscript is well redacted and explain very well the purpose of the research.

The manuscript can be published in the present form.

Congratulate the authors for the review performed.

Abstract: Is well exposed and resume the major points of the review.

Introduction: Is well structured and explain how nowadays the chronic low back pain is in the population and the different causes that produced based in the previous literature.

Material and Methods: Are well structured and explained.

Results: Are well exposed and explained

Conclusions. Are adapted to the review findings

Reviewer 4 Report

The paper is well written and deeply analyze the review topic. I have only few suggestion:

- The keywors used to search the papers for this topic are not provided. Plase add them.

- When the data recording was conduceted?
